# Congenital Heart Defects in Patients with Molecularly Confirmed Sotos Syndrome

**DOI:** 10.3390/diagnostics14060594

**Published:** 2024-03-11

**Authors:** Giulio Calcagni, Federica Ferrigno, Alessio Franceschini, Maria Lisa Dentici, Rossella Capolino, Lorenzo Sinibaldi, Chiara Minotti, Alessia Micalizzi, Viola Alesi, Antonio Novelli, Anwar Baban, Giovanni Parlapiano, Domenico Coviello, Paolo Versacci, Carolina Putotto, Marcello Chinali, Fabrizio Drago, Andrea Bartuli, Bruno Marino, Maria Cristina Digilio

**Affiliations:** 1Area of Fetal, Neonatal, and Cardiological Sciences, Bambino Gesù Children’s Hospital, IRCCS, 00146 Rome, Italy; federicaferrigno94@gmail.com (F.F.); alessio.franceschini@opbg.net (A.F.); marcello.chinali@opbg.net (M.C.); fabrizio.drago@opbg.net (F.D.); 2The School of Pediatrics, University of Rome “Tor Vergata”, 00173 Rome, Italy; 3Medical Genetics, Translational Pediatrics and Clinical Genetics Research Area, Bambino Gesù Children’s Hospital, IRCCS, 00146 Rome, Italy; marialisa.dentici@opbg.net (M.L.D.); rossella.capolino@opbg.net (R.C.); lorenzo.sinibaldi@opbg.net (L.S.); mcristina.digilio@opbg.net (M.C.D.); 4Medical Genetics Section, Department of Biomedicine and Prevention, University of Rome “Tor Vergata”, 00173 Rome, Italy; chiara.minotti95@gmail.com; 5Laboratory of Medical Genetics, Translational Cytogenomics Research Unit, Bambino Gesù Children’s Hospital, IRCCS, 00146 Rome, Italy; alessia.micalizzi@opbg.net (A.M.); viola.alesi@opbg.net (V.A.); antonio.novelli@opbg.net (A.N.); 6Cardiogenetic Center, Bambino Gesù Children’s Hospital, IRCCS, 00146 Rome, Italy; anwar.baban@opbg.net (A.B.); giovanni.parlapiano@opbg.net (G.P.); 7Laboratory of Human Genetics, Istituto Giannina Gaslini, IRCCS, 16147 Genoa, Italy; domenicocoviello@gaslini.org; 8Department of Maternal Infantile and Urological Sciences, Sapienza University of Rome, 00161 Rome, Italy; paolo.versacci@uniroma1.it (P.V.); carolina.putotto@uniroma1.it (C.P.); bruno.marino@uniroma1.it (B.M.); 9Rare Diseases and Medical Genetics, Bambino Gesù Children’s Hospital, IRCCS, 00146 Rome, Italy; andrea.bartuli@opbg.net

**Keywords:** sotos syndrome, congenital heart defect, cardiomyopathy, *NSD1* gene

## Abstract

Sotos syndrome is an autosomal dominant condition characterized by overgrowth with advanced bone age, macrodolicocephaly, motor developmental delays and learning difficulties, and characteristic facial features caused by heterozygous pathogenetic variants in the *NSD1* gene located on chromosome 5q35. The prevalence of heart defects (HDs) in individuals with Sotos syndrome is estimated to be around 15–40%. Septal defects and patent ductus arteriosus are the most commonly diagnosed malformations, but complex defects have also been reported. The aim of our study was to analyze the prevalence of HD, the anatomic types, and the genetic characteristics of 45 patients with Sotos syndrome carrying pathogenetic variants of *NSD1* or a 5q35 deletion encompassing *NSD1*, who were followed at Bambino Gesù Children’s Hospital in Rome. Thirty-nine of the forty-five patients (86.7%) had a mutation in *NSD1*, while six of the forty-five (13.3%) had a deletion. Most of the patients (62.2%, 28/45) were male, with a mean age of 14 ± 7 years (range 0.2–37 years). A total of 27/45 (60.0%) of the patients had heart defects, isolated or combined with other defects, including septal defects (12 patients), aortic anomalies (9 patients), mitral valve and/or tricuspid valve dysplasia/insufficiency (1 patient), patent ductus arteriosus (3 patients), left ventricular non-compaction/hypertrabeculated left ventricle (LV) (4 patients), aortic coarctation (1 patient), aortopulmonary window (1 patient), and pulmonary valve anomalies (3 patients). The prevalences of HD in the two subgroups (deletion versus intragenic mutation) were similar (66.7% (4/6) in the deletion group versus 58.91% (23/39) in the intragenic variant group). Our results showed a higher prevalence of HD in patients with Sotos syndrome in comparison to that described in the literature, with similar distributions of patients with mutated and deleted genes. An accurate and detailed echocardiogram should be performed in patients with Sotos syndrome at diagnosis, and a specific cardiological follow-up program is needed.

## 1. Introduction

Sotos syndrome (OMIM #117550) is an autosomal dominant condition characterized by overgrowth with advanced bone age, macrodolicocephaly, motor developmental delay and learning difficulties, and characteristic facial features. It was first described by Sotos et al., who reported, in 1964, five patients with some typical features of the syndrome [1,2], which has an estimated incidence of 1 in 14,000 live births. Heterozygous variants of the *NSD1* gene encoding a histone methyltransferase located on chromosome 5q35 have been identified as the cause of Sotos syndrome [3]. Patients with the Sotos syndrome phenotype carrying a 5q35 deletion encompassing the *NSD1* gene have also been reported in about 5–20% of Caucasian patients [3,4,5,6]. Interestingly, submicroscopic deletions involving the entire *NSD1* gene are the major causes of Sotos syndrome in Japanese patients, accounting for more than 50% of the subjects [7,8].

The *NSD1* gene (MIM 606681) is mapped to 5q35.2–q35.3 and consists of 23 exons, the first of which is noncoding [9]. To date, over 400 different mutations in *NSD1* associated with Sotos syndrome have been reported. As for the role of the *NSD1* gene, it was initially identified as encoding a nuclear receptor-interacting protein and potential transcriptional cofactor; however, its direct involvement in gene transcription remains unclear [10,11].

Although not among the most common features of the syndrome, cardiac complications have been extensively studied in this pathology.

The prevalence of congenital heart defects (CHDs) in individuals with Sotos syndrome has been estimated to be around 15–25% in several studies [1,12,13,14], although other studies have reported a higher prevalence of around 40% [6]. Septal defect and patent ductus arteriosus (PDA) are the most commonly diagnosed malformations, but complex defects have also been reported, including tricuspid and pulmonary atresia [15], aortopulmonary window, pulmonary valve (PV) stenosis, and tetralogy of Fallot [16]. A high prevalence of Left Ventricular Non-Compaction (LVNC) has been observed [17,18,19].

In this study, we analyzed the prevalence of cardiac anomalies, the anatomic types, and the genetic characteristics of 45 patients with Sotos syndrome carrying pathogenetic variants of *NSD1* or a 5q35 deletion encompassing *NSD1*. All patients had access to care at our third-level center.

## 2. Methods

Between June 2011 and December 2022, 39 patients (16 females, 23 males) with Sotos syndrome carrying a pathogenetic variant of *NSD1* gene and 6 patients (5 males, 1 female) with a 5q35 deletion were evaluated at our hospitals. The mean age at the time of last cardiological evaluation was 14 ± 7 years (range 0.2–37 years).

Genomic DNA was extracted from circulating leukocytes using the QIAampH DNA Blood Kit (QIAGEN Sciences, Germantown, MD, USA) according to the manufacturer’s instructions.

Mutational analyses of the NSD1 gene (NCBI reference sequences NG_032003.1, NM_013275.5, and NP_037407.4) were performed using customized panels, SeqCap EZ Choice Enrichment Kit (Roche, Madison, WI, USA), or the Twist Human Clinical Exome Panel (Twist Bioscience South San Francisco, CA, USA) on NextSeq550 or NovaSeq6000 sequencing platforms (Illumina, San Diego, CA, USA) according to the manufacturer’s protocol. Sequences were aligned to the human genome build UCSC GRCh37 using DRAGEN Germline Pipeline of BaseSpace (Illumina, https://basespace.illumina.com/, accessed on 22 January 2024) and analyzed for variant prioritization and annotation using Geneyx analysis software (LifeMap Sciences, Inc., Covina, CA, USA, v5.16 Build 2305). All suspected variants were checked in public databases (dbSNP, Exome Aggregation Consortium (ExAC), and Genome Aggregation Database (GnomAD)) and filtered for allele frequency or MAF < 1% in GnomAD, species conservation of the underlying amino acid, and a change in the protein’s primary structure (the variants were evaluated using VarSome [20] and categorized in accordance with the American College of Medical Genetics (ACMG) recommendations [21]. Variants were analyzed for coverage and Qscores (minimum threshold of 30) and visualized using the Integrative Genome Viewer (IGV) [22]. Clinically relevant single nucleotide variants (SNVs) in index cases were validated in re-extracted DNA and verified in available family members using bidirectional Sanger sequencing.

Standard karyotypes were generated for all patients. Microarray analysis was conducted at a practical average resolution of 100 kb using Array-CGH (4 × 180 K oligo array, Agilent) or SNP array (Beadchip 850K, Illumina) platforms. In three patients, specific MLPA probes for the NSD1 gene were used [6].

Array-CGH and SNP images were obtained using the Agilent DNA Microarray Scanner and Agilent Scan Control Software (v A.8.4.1) and data were analyzed using Agilent CytoGenomics (v 4.0.3.12) or Illumina BlueFuse Multi (v 4.4), depending on the platform. Confirmation and parental segregation analysis were performed using real-time PCR of the NSD1 gene using a SYBR Green assay [23]. Informed consent was obtained from the patients’ parents. The study was conducted in accordance with the Helsinki Declaration. Considering the retrospective nature of the analysis, the current study did not require the approval of the local ethics committee according to current legislation. Data were retrospectively analyzed in accordance with personal data protection policies.

The criteria for inclusion in the study were a confirmed molecular diagnosis of Sotos syndrome and the availability of at least one complete echocardiographic examination for each patient. All patients were evaluated by pediatric cardiologists and clinical geneticists. Systematic patient interviews investigated family and personal history of genetic syndromes, metabolic disorders, known congenital heart defects, and previous cardiac surgery procedures. Cardiac evaluation consisted of chest X-rays, electrocardiograms, and two-dimensional color Doppler echocardiography for all patients. An Epiq7 ultrasound machine (Philips Medical, Andover, MA, USA) with a 1–5 MHz transducer (X5-1) was used for studying all patients. Conventional echocardiographic evaluations were performed according to the recommendations of the American Society of Echocardiography [24]

Physiological/trivial mitral, tricuspid, or aortic valve insufficiencies were excluded, particularly where valves have echocardiographically normal anatomies. Similarly, we considered non-compacted or hypertrabeculated LV based on the medical literature [25].

## 3. Results

We studied 45 patients with a clinical diagnosis of Sotos syndrome, confirmed using molecular analysis, including 39 patients carrying a pathogenetic variant of the NSD1 gene (Group 1) and 6 patients with a 5q35 deletion (Group 2).

The mutations identified in the present study are indicated in detail in Figure 1.

### 3.1. Patients with NSD1 Variants (Group 1)

Group 1 was the largest in our population (39/45 = 86.6%). CHD was diagnosed in 23/39 (58.9%) of the patients (11 females and 12 males). Various types of cardiac anomalies were found in these patients, including atrial septal defect (ASD), ventricular septal defect (VSD) (4 muscular ventricular septal defects, 2 perimembranous septal defects), aortic anomalies (coarctation, bicuspid, dysplasia, dilatation, or stenosis), (PDA), non-compacted and hypertrabeculated LV, aortopulmonary window, mitral, tricuspid, and pulmonary valve anomalies. The heart phenotype and the associated genotype are shown in Table 1 and Table 2.

The prevalence of cardiac defects in the single group of variants is shown in Table 3, revealing that the occurrence of cardiac anomalies is only mildly overrepresented in patients with missense variants.

One affected patient, who was the father of a child with a c.6455G>A (p.Arg2152Gln) mutation and CHD, was of tall stature and had mild learning difficulties, macrocephaly, facial anomalies (triangular face, high forehead, down-slanting palpebral fissures, pointed chin), contractures on both elbows and kyphosis.

### 3.2. Patients with 5q Deletion (Group 2)

CHD was diagnosed in 4/6 (66.7%) of the patients (all males). Anatomic types included ASD, PDA, hypertrophy, and hypertrabeculation of the LV.

Cardiac anatomy and molecular data of patients with 5q deletions are listed in Table 4.

### 3.3. Extracardiac Anomalies

All patients in our cohort were described in detail, and their specific extracardiac features and *NSD1* variants are given in Table 5.

## 4. Discussion

The prevalence of CHD in our patients with pathogenetic variants in the *NSD1* gene is 58.9%, while the literature reports a prevalence ranging from 15 to 40% [10,13,26].

Although there is no selection bias, since we enrolled all patients undergoing genetic analysis at our hospital, the elevated prevalence of heart defects in our patient population might be attributable to our center’s specialization in cardiology and cardiac surgery. It is possible that our facility’s expertise influenced some patients, previously monitored at other national centers, to opt for continued follow-up at our institution for enhanced cardiological care. At the same time, our center has expertise in the fields of genetics and neuropsychiatry, making it a reference point for all those patients with Sotos syndrome who do not have cardiovascular anomalies.

The spectrum of abnormalities identified in our patients encompasses a wide range, extending from relatively straightforward defects like atrial or ventricular septal defects to more complex forms of heart disease.

In Table 6, we report the prevalence of cardiac defects in patients with Sotos syndrome and *NSD1* variants in the literature.

### 4.1. Septal Defects

The predominant cardiac anomalies observed in the current study are septal defects, encompassing both atrial ostium secundum (ASDos) and ventricular (VSD) variations. Septal defects, whether occurring as isolated anomalies or in conjunction with cardiac valvular defects, constitute a significant portion of the cases. Specifically, among the VSD cases, four manifested as muscular septal defects, while two presented as perimembranous defects. Notably, only two patients underwent cardiac surgery for ASD closure, and none required surgical intervention for VSD closure.

### 4.2. Aortic Anomalies

Aortic anomalies were the second most prevalent defects in our cohort. Half of the patients with aortic defects were diagnosed with BAV, alone or associated with aortic coarctation or supravalvular aortic stenosis. Aortic valve dysplasia characterized by thickening of the flaps was observed in additional patients in this group.

Ascending aorta dilation was also diagnosed in four patients in our cohort and is frequently documented to occur in association with Sotos syndrome [2,28,29]. It can be isolated or associated with various types of valvular anomalies. The prevalence of aortic dilation in our population is comparable to that of Foster’s study [2]. These authors concluded that this defect, which is identified in patients with Sotos syndrome, as described in the literature, is of limited clinical interest because the defect does not progress with growth. The presence of this feature in patients with Sotos syndrome may probably be considered a specific “cardiac overgrowth sign”, even though the absence of progression over time may refute this hypothesis. However, it is important to note that in our population, all patients with aortic dilation exhibited a more complex phenotype, requiring prompt follow-up and, in some cases, the use of medical therapy. It would be beneficial to gather larger samples of patients with Sotos syndrome and aortic anomalies to assess the extent to which the issue is either benign or warrants attention.

### 4.3. Mitral and Tricuspid Valve Anomalies

Mitral and/or tricuspid valve dysplasia was a prevalent occurrence in our patient population, manifesting either as an independent anomaly or in association with other defects. The anatomical descriptions consistently revealed moderate mitral or tricuspid insufficiency in cases where these dysplasias were present.

### 4.4. LVNC

Hypertrabeculation of the LV affected four individuals in the present study, manifesting either as isolated LVNC or associated with other cardiac defects. The criteria used to define LVNC, according to the literature, were met by two patients [25]. One patient presented with moderate LV dysfunction requiring medical therapy. In the available literature, four unrelated cases documented left ventricular non-compaction in patients with Sotos syndrome, with one case also being associated with a double orifice mitral valve [17,18,19].

### 4.5. PDA

Three patients exhibited a patent ductus arteriosus. Therefore, the percentage of patients with PDA in our study was lower than that of patients with septal defects, whereas, in most studies, it is comparable to that of septal defects [12,14,16].

### 4.6. Aortopulmonary Window

Interestingly, we found a patient with aortopulmonary window, a rare heart disease that has previously been associated with Sotos syndrome in a single case [16]. This medical condition represents 0.2–0.6% of all congenital malformations and, to date, has not been linked to a specific genetic anomaly or syndrome [30]. Considering that the genetic basis for aortopulmonary window is little known, this specific association may point to *NSD1* being the causative gene for this cardiac defect.

### 4.7. Genotype–Phenotype Correlations

Another crucial aspect to analyze is the correlation between genotypes and phenotypes when it comes to intragenic pathogenic variants and 5q35 microdeletions. It is important to note, based on epidemiological studies, that intragenic mutations, rather than submicroscopic deletions, are the predominant cause of Sotos syndrome in non-Japanese patients. In non-Japanese populations, 5q35 microdeletions are uncommon, accounting for only 10% of affected individuals [5,10,31]. This trend is consistent in our cohort as well, where we identified 39 patients with gene variants compared to only 6 patients with a deletion.

From the first cohort studies, a consistent distinction in the incidence of heart disease emerged between patients with point mutations and those with deletions. Previously, this disparity was attributed to a potential role of a gene neighboring *NSD1*, as *NSD1* itself did not appear to be expressed in the heart [16]. Cole et al. initially documented a notable prevalence of diverse CHDs and urogenital anomalies in their Japanese reviews in 1990, hinting at the potential existence of a distinct syndrome in Japanese patients. However, clinical manifestations and follow-up data clearly showed that these Japanese patients had typical Sotos syndrome [32,33].

In 2003, Kurotaki et al. [8] observed a higher prevalence of congenital heart anomalies in patients with microdeletions than those with *NSD1* point mutations. Similar findings were reported by Nagai et al. in 2003 and Saugier-Veber et al. in 2007 [27,34]. Moreover, in 2005, Tatton-Brown and colleagues, analyzing a population of 266 individuals with a genetic alteration in *NSD1*, found that individuals with 5q35 mutations tended towards more cardiac anomalies than individuals with deletions, similar to reports by Cecconi et al. [6,13].

Our cohort displayed a similar prevalence of heart disease in patients with point mutations and deletions (58.9 vs. 66.7%). Nevertheless, the most complex heart diseases were identified in patients with mutations potentially linked to the larger number of mutation cases in our cohort.

Our cohort contained non-related patients, except for two who were father and son. Interestingly, these two patients carried the same mutation but exhibited different cardiac phenotypes. Specifically, the father had a normal heart, while the son presented with a complex heart disease characterized by aortic coarctation, mitral and aortic dysplasia, and tricuspid regurgitation. This aligns with the concept that a single mutation can manifest diverse phenotypic expressions.

In this regard, no hotspot mutations for cardiac defects were identified. Different single variants were identified in the present study. The only variant that was common in two patients is c.5740C>T, which was identified in two unrelated patients with cardiac defects who presented with different heart malformations: atrial septal defect ostium secundum type in one patient and aortic dilatation in the second patient. The two splicing variants involving the 5622 residues were detected in patients with discordant phenotypes (LVNC with mitral and tricuspid anomaly in one; normal heart in the other).

Nonsense, frameshift, and missense variants are all associated with cardiac defects. As shown, the prevalence of cardiac defects in the single group of variants confirms that the occurrence of cardiac anomalies is only mildly overrepresented in patients with missense variants.

## 5. Conclusions

Heart disease in patients with Sotos syndrome cannot be underestimated. The prevalence of heart disease was notably high in our cohort, affecting more than half of the patients, with similar distribution among patients with mutated and deleted genes. Anatomic malformations include septal defects, aortic anomalies, mitral, pulmonary, and tricuspid defects, and mitral valve and/or tricuspid valve anomalies. A higher prevalence of LVNC than previously reported was documented in the present study. The association with the rare cardiac defect, aortopulmonary window, was noted. Aortic dilatation, either isolated or in association with other defects, has been confirmed as a clinical feature detectable in patients with Sotos syndrome.

Based on these observations, an accurate and detailed cardiac evaluation using an echocardiogram should be performed in all patients with Sotos syndrome at diagnosis. A specific cardiological follow-up program is also needed. Furthermore, it appears essential to continue research on patients with Sotos syndrome to study individual mutations and highlight potential links with specific congenital heart conditions.

## Figures and Tables

**Figure 1 diagnostics-14-00594-f001:**
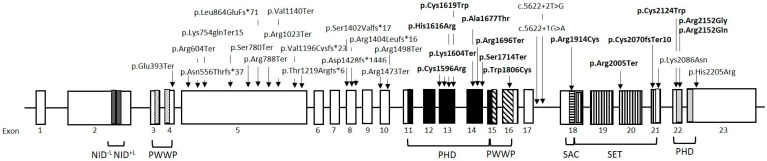
Schematic representation of the NSD1 gene. Distinct boxes represent different functional domains: two nuclear receptor interacting domains (NIDs), two proline–tryptophan–tryptophan–proline (PWWP), five plant homodomain zinc fingers (PHDs), and one SET domain and SAC domain. The mutations identified in the present study are indicated with arrows. The variants in the domains are in bold.

**Table 1 diagnostics-14-00594-t001:** Cardiac and molecular details of patients with Sotos syndrome and *NSD1* variants.

NSD1 Variant	Type of Mutation	Segregation	Heart Defects
c.1177G>T (p.Glu393Ter)	nonsense	paternal	- BAV with moderate AoI - MV dysplasia with moderate MI
c.1667_1685del (p.Asn556ThrfsTer37)	frameshift	parents not tested	- TV dysplasia with moderate TI
c.1810C>T (p.Arg604Ter)	nonsense	parents not tested	- Normal heart
c.2258_2259insTC p.Lys754GlnfsTer15	frameshift	de novo	- Supravalvular aortic stenosis - Mild dilation of AAo - BAV with moderate AoI
c.2339C>A (p.Ser780Ter)	nonsense	de novo	- ASDos - VSD (perimembranous) - Dysplastic PV with stenosis
c.2362C>T (p.Arg788Ter)	nonsense	de novo	- Normal heart
c.2591_2599 delinsGTAAGGA (p.Leu864CysfsTer10)	frameshift	de novo	- VSD (muscular)
c.2906delG (p.Gly969GlufsTer71)	frameshift	de novo	- PV stenosis - Moderate MI - Mild dilation of AAo
c.3067C>T (p.Arg1023Ter)	nonsense	parents not tested	- Normal heart
c.3091C>T (p.Arg1031Ter)	nonsense	de novo	- Normal heart
c.3418delGT	frameshift	de novo	- Normal heart
c.3585delT (p.Val1196CysfsTer23)	frameshift	de novo	- Normal heart
c.3656_3657del (p.Thr1219ArgfsTer6)	frameshift	parents not tested	- AoV dysplasia with moderate AoI
c.4204del (p.Ser1402ValfsTer17)	frameshift	parents not tested	- PDA
c.4210_4211insTT (p.Arg1404LeufsTer16)	frameshift	parents not tested	- Normal heart
c.4287insGG (p.Asp1428fsTer1446)	nonsense	de novo	- Normal heart
c.4417C>T (p.Arg1473Ter)	nonsense	parents not tested	- ASDos - VSD (perimembranous)
c.4493C>G (p.Ser1498Ter)	nonsense	parents not tested	- PDA - MV dysplasia with moderate MI
c.4786T>C (p.Cys1596Arg)	missense	de novo	- Mild dilation of AAo - LVH
c.4810A>T (p.Lys1604Ter)	nonsense	de novo	- Normal heart
c.4847A>G (p.His1616Arg)	missense	de novo	- ASDos - VSD (muscular)
c.4857T>G (p.Cys1619Trp)	missense	de novo	- VSD (muscular) - Dysplastic PV with stenosis - Hypertrabeculated LV
c.5029G>A (p.Ala1677Thr)	missense	de novo	- MV dysplasia with moderate MI - TV dysplasia with moderate TI - ASD - VSD (muscular) - Moderate AoI
c.5084dupC (p.Arg1696Ter)	frameshift	de novo	- Normal heart
c.5141C>G (p.Ser1714Ter)	nonsense	de novo	- AoV dysplasia with moderate AoI - MV dysplasia with mild MI
c.5418G>T (p.Trp1806Cys)	missense	de novo	- Normal heart
c.5581C>T (p.Arg1861Ter)	nonsense	de novo	- ASDos
c.5622+2T>G	splicing	de novo	- Hypertrabeculated LV - Moderate MI
c.5622+1G>A	splicing	de novo	- Normal heart
c.5740C>T (p.Arg1914Cys)	missense	de novo	- ASDos
c.5740C>T (p.Arg1914Cys)	missense	de novo	- Mild dilatation of AAo
c.6204_6213delTGTTTGCAAA (p.Cys2070fsTer10)	frameshift	de novo	- Normal heart
c.6013C>T (p.Arg2005Ter)	nonsense	de novo	- Aortopulmonary window
c.6258G>C (p.Lys2086Asn)	missense	de novo	- Normal heart
c.6372T>G (p.Cys2124Trp)	missense	de novo	- LVNC
c.6454C>G (p.Arg2152Gly)	missense	de novo	- Normal heart
c.6455G>A (p.Arg2152Gln)	missense	paternal	- Aortic coarctation - MV dysplasia with moderate MI - AoV dysplasia with moderate AoI
c.6455G>A (p.Arg2152Gln)	missense	father of the previous patient	- Normal heart
c.6614A>G (p.His2205Arg)	missense	de novo	- LVNC - Moderate LV dysfunction

Abbreviations: AAo: ascending aorta; AoI: aortic insufficiency; AoV: aortic valve; ASD: atrial septal defect; ASDos: atrial septal defects ostium secundum; BAV: bicuspid aortic valve; LV: left ventricle; LVH: left ventricle hypertropia; LVNC: left ventricular non-compaction; MI: mitral insufficiency; MV: mitral valve; PDA: patent ductus arteriosus; PV: pulmonary valve; TI: tricuspid insufficiency; TV: tricuspid valve; VSD: ventricular septal defect.

**Table 2 diagnostics-14-00594-t002:** Review of anatomic and frequency characteristics of the different types of CHD in patients with *NSD1* variants.

Cardiac Defect	Number
**Septal defects**	
- ASDos	2
- ASDos + VSD (1 pts perimembranous VSD, 1 pt muscular VSD)	2
- ASDos + VSD perimembranous + dysplastic and stenotic PV	1
- ASDos + VSD + dysplastic MV and TV, AoI	1
- VSD muscular	1
- VSD + Dysplastic PV + Hypertrabeculated LV	1
**Aortic anomalies**	
- AAo dilation	2
- Aortic coarctation + MV and AoV dysplasia + TI	1
- BAV + AoV, MV dysplasia with MI	1
- Supravalvular aortic stenosis + BAV + AAo dilatation	1
- AoV dysplasia and AoI	1
- AoV + MI	1
**Mitral valve and Tricuspid valve anomalies**	
- TI	1
- MI + hypertrabeculated LV	1
**Patent ductus arteriosus**	
- PDA	1
- PDA + MI	1
**Left ventricular non-compaction**	
- LVNC	2
**Aortopulmonary window**	
- Aortopulmonary window	1
**Pulmonary valve anomalies**	
- PV stenosis + MI + AAo dilation	1

Abbreviations: AAo: ascending aorta; AoI: aortic insufficiency; AoV: aortic valve; ASDos: atrial septal defects ostium secundum; BAV: bicuspid aortic valve; LV: left ventricle; LVNC: left ventricular non-compaction; MI: mitral insufficiency; MV: mitral valve; PDA: patent ductus arteriosus; PV: pulmonary valve; TI: tricuspid insufficiency; TV: tricuspid valve; VSD: ventricular septal defect.

**Table 3 diagnostics-14-00594-t003:** Prevalence of congenital heart defects in patients with nonsense, frameshift, splicing, and missense variants.

Type of Variant	with CHD (Number)	%	Normal Heart	%
**Nonsense/Frameshift/Splicing**	**14/26**	**53.8**	**12/26**	**46.2**
- Nonsense	7/13	53.8	6/13	46.2
- Frameshift	6/11	54.5	5/11	45.5
Splicing	1/2	50	1/2	50
**Missense**	**9/13**	**69.2**	**4/13**	**30.8**

**Table 4 diagnostics-14-00594-t004:** Cardiac and molecular details of patients with Sotos syndrome and *NSD1* deletions.

*NSD1* Deletion	OMIM Genes	Segregation	Heart Defects
*NSD1* deletion (MLPA)		de novo	- Normal heart
arr[GRCh37] 5q35.3(176681801_176697024)x1 (15.2 kb)	*NSD1*	de novo	- ASDos
*NSD1* deletion (MLPA)		de novo	- mild LVH
arr[GRCh37] del 5q35.2q35.3 (175347741-177587471)x1 (2.24 Mb)	27 OMIM genes; 9 OMIM disease-causing genes (*SNCB*, *FGFR4*, ***NSD1***, *SLC34A1*, *F12*, *DDX41*, *B4GALT7*, *PROP1*, *NHP2)*	de novo	- Normal heart
Arr[GRCh37 ]del5q35.2q35.3 (176378453-176735244)x1 (356.8 kb)	5 OMIM genes; 2 OMIM disease-causing genes (*FGFR4*,***NSD1***)	parents not tested	- Hypertrabeculated LV
*NSD1* deletion (MLPA)		parents not tested	- PDA

Abbreviations: ASDos: atrial septal defects ostium secundum; LV: left ventricle; LVH: left ventricle hypertropia; LVNC: left ventricular non-compaction; PDA: patent ductus arteriosus.

**Table 5 diagnostics-14-00594-t005:** Extracardiac features of patients with Sotos syndrome in the present study.

Patient	Developmental Delay	Epilepsy	Tall Stature (>97th Percentile)	Macrocephaly (>97th Percentile)	Advanced Bone Age (>2 Years)	Renal Anomaly	Skeletal Anomaly	*NSD1* Variant
1	+	-	+	+	+	-	-	c.1177G>T
2	+	-	+	+	+	-	+	c.1667_1685del
3	+	+	+	+	+	-	-	c.1810C>T
4	+	-	+	+	+	-	+	c.2258_2259insTC
5	+	-	+	+	+	+	+	c.2339C>A
6	+	+	-	+	-	-	+	c.2362C>T
7	+	+	+	+	+	+	-	c. 2591_2599 delinsGTAAGGA
8	+	-	+	+	-	+	-	c.2906delG
9	+	+	+	-	-	-	-	c.3067C>T
10	+	-	+	+	+	-	-	c.3091C>T
11	+	-	+	+	+	-	-	c.3418delGT
12	+	-	+	+	+	-	+	c.3585delT
13	+	-	+	+	-	-	+	c.3656_3657del
14	+	+	-	+	-	-	+	c.4204del
15	+	-	+	+	+	+	-	c.4210_4211insTT
16	+	-	+	+	+	-	-	c.4287insGG
17	+	-	+	+	+	-	-	c.4417C>T
18	+	+	+	+	+	-	-	c.4493C>G
19	+	-	-	+	-	-	+	c.4786T>C
20	+	-	+	+	+	-	-	c.4810A>T
21	+	+	+	+	+	-	+	c.4847A>G
22	+	-	+	+	+	+	-	c.4857T>G
23	+	-	+	+	+	-	+	c.5029G>A
24	+	-	+	-	+	-	-	c.5084dupC
25	+	+	+	+	+	-	+	c.5141C>G
26	+	+	+	+	--	-	-	c.5418G>T
27	+	-	+	+	+	-	-	c.5581C>T
28	+	-	+	+	+	-	+	c.5622+2T>G
29	+	-	+	+	+	-	-	c.5622+1G>A
30	+	+	-	+	-	-	-	c.5740C>T
31	+	-	+	+	+	-	-	c.5740C>T
32	+	-	+	+	+	-	+	c.6204_6213delTGTTTGCAAA
33	+	-	+	+	+	+	-	c.6013C>T
34	+	-	+	+	+	-	-	c.6258G>C
35	+	-	+	+	+	-	+	c.6372T>G
36	+	+	+	+	+	-	+	c.6454C>G
37	+	-	+	+	+	-	+	c.6455G>A
38	+	-	+	+	NK	-	+	c.6455G>A
39	+	-	+	+	+	-	-	c.6614A>G
40	+	-	-	+	-	-	+	NSD1 deletion (MLPA)
41	+	+	+	+	+	-	+	5q35.3(176681801_176697024)x1
42	+	-	+	+	-	-	-	NSD1 deletion (MLPA)
43	+	-	-	+	+	+	+	del 5q35.2q35.3 (175347741-177587471)x1
44	+	+	+	+	+	-	-	del5q35.2q35.3 (176378453-176735244)x1
45	+	-	+	+	-	-	+	NSD1 deletion (MLPA)

**Table 6 diagnostics-14-00594-t006:** Prevalence of cardiac defects in patients with Sotos syndrome and *NSD1* variants in the literature.

References [4,6,10,13,26,27]	Prevalence of Cardiac Defects
Rio et al., 2003	9.7%
Tatton-Brown et al., 2005	21%
Cecconi et al., 2005	43.7%
Saugier-Veber et al., 2007	8%
Ha et al., 2016	33.3%
Lourdes et al., 2023	38.7%

## Data Availability

Additional data are available from the corresponding author upon reasonable request.

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
