# Peer review of "Congenital Heart Defects in Patients with Molecularly Confirmed Sotos Syndrome"

_diagnostics, 2024, doi:10.3390/diagnostics14060594_

Round 1

Reviewer 1 Report

Comments and Suggestions for Authors

The article "Congenital heart defects in patients with molecularly confirmed Sotos syndrome," authored by Calcagni and colleagues, is an interesting piece of work related to describing heart defects in this hypergrowth syndrome. While this study presents compelling insights and valuable data for the field, some additions would enhance the relevance of the work.

1.-It would be valuable to separate in a new table those variants related to missense from those related to non-sense or frameshift mutations. Describe and discuss more about the variants found, especially as these are novel and have not been previously described in populations related to Sotos patients.

2.-The authors should add an image related to protein domains of NSD1 to enhance the discussion of missense variants related to this syndrome, particularly those described heart defects not previously displayed.

3.As Sotos syndrome is related to complete penetrance, add a description of the father of the patient who displayed a paternally linked variant c.1177G>T (p.Glu393Ter), for example, overgrowth as described previously by Lacetta and colleagues in 2017 (DOI: 10.3389/fped.2017.00236).

4.-As cardiovascular signs previously reported in Sotos syndrome are essentially congenital heart defects such as septal defects, patent ductus arteriosus, and pulmonary stenosis, etc., the authors found almost 20% of aortic anomalies, beyond just aortic dilation. Discuss this finding further.

Comments on the Quality of English Language

Some typos were found, and grammar requires careful revision. For example, "Defects and" instead of "Defect sand” Line 69.

In the methods section, some sentences are poorly integrated, as seen in line 117: "The inclusion criteria for the study were confirmed molecular diagnosis of Sotos syndrome, ability to perform a complete echocardiographic exam"; please review the entire text for coherence.

Author Response

REVIEWER #1:

The article "Congenital heart defects in patients with molecularly confirmed Sotos syndrome," authored by Calcagni and colleagues, is an interesting piece of work related to describing heart defects in this hypergrowth syndrome. While this study presents compelling insights and valuable data for the field, some additions would enhance the relevance of the work.

1.-It would be valuable to separate in a new table those variants related to missense from those related to non-sense or frameshift mutations. Describe and discuss more about the variants found, especially as these are novel and have not been previously described in populations related to Sotos patients.

We thank the reviewer for this valuable suggestion. We added a new table in the manuscript with the prevalence of congenital heart defects for each specific mutation (nonsense, frameshift, splicing and missense variants). Few sentence have been also added in the discussion.

2.-The authors should add an image related to protein domains of NSD1 to enhance the discussion of missense variants related to this syndrome, particularly those described heart defects not previously displayed.

We thank the reviewer for this suggestion. We added a figure (fig1) in the text with a schematic representation of the NSD1 gene. Particularly, the mutations identified in the present study are indicated with arrow in the same figure.

3. As Sotos syndrome is related to complete penetrance, add a description of the father of the patient who displayed a paternally linked variant c.1177G>T (p.Glu393Ter), for example, overgrowth as described previously by Lacetta and colleagues in 2017 (DOI: 10.3389/fped.2017.00236).

We thank the reviewer for this comment. As suggested we add in the results a description of clinical characteristics the father of our patient.

4. As cardiovascular signs previously reported in Sotos syndrome are essentially congenital heart defects such as septal defects, patent ductus arteriosus, and pulmonary stenosis, etc., the authors found almost 20% of aortic anomalies, beyond just aortic dilation. Discuss this finding further.

We thank the reviewer for this comment. The involvement of the aortic anomalies in Sotos Syndrome has been previously described and we discussed our data  in comparison of the literature. Probably, it should be considered as a “cardiac sign” of this overgrowth syndrome, even though no progression in the aortic diameter has been reported. We added a comment in the discussion as required.

  1. Some typos were found, and grammar requires careful revision. For example, "Defects and" instead of "Defect sand” Line 69.

 We corrected the text accordingly.

In the methods section, some sentences are poorly integrated, as seen in line 117: "The inclusion criteria for the study were confirmed molecular diagnosis of Sotos syndrome, ability to perform a complete echocardiographic exam"; please review the entire text for coherence.

We modified the text accordingly with the suggestion. Thanks.

Reviewer 2 Report

Comments and Suggestions for Authors

This is a good piece of work on Sotos syndrome with the required analysis done. Some grammatical and typo errors need to be taken care.

For example, page 3 2nd paragraph - Standard chromosome...

Comments on the Quality of English Language

Minor changes in English required as suggested above. 

Author Response

REVIEWER #2

This is a good piece of work on Sotos syndrome with the required analysis done. Some grammatical and typo errors need to be taken care.

For example, page 3 2nd paragraph - Standard chromosome...

Minor changes in English required as suggested above. 

We thank the reviewer for the appreciable comment to our manuscript. We corrected sentences in the text according to the suggestion.

Reviewer 3 Report

Comments and Suggestions for Authors

The presented article «Congenital heart defects in patients with molecularly con-firmed Sotos syndrome» is certainly relevant for the modern medicine. Given the low incidence of Sotos syndrome, the number of children recruited to the analysis is sufficient. At the same time, some minor issues must be solved:

1. The tables should be presented in a different format: now they are readable, but not very presentable.

2. Please include in the manuscript a table with a clinical description and concomitant diseases of the studied children.

3. The list of references contains a lot of sources older than 5 years.

Comments on the Quality of English Language

Moderate editing of English language required

Author Response

The presented article «Congenital heart defects in patients with molecularly con-2 firmed Sotos syndrome» is certainly relevant for the modern medicine. Given the low incidence of Sotos syndrome, the number of children recruited to the analysis is sufficient. At the same time, some minor issues must be solved:

  1. The tables should be presented in a different format: now they are readable, but not very presentable.

We thank the reviewer for this comment. We agree that even though our tables are well readable and exhaustive, they are also not very presentable. We suggested in a specific comment added in the text, to remove one column (gender) in table 1 and table 4. In this way it could be easily for the editorial office, to layout them in a better way.   

  1. Please include in the manuscript a table with a clinical description and concomitant diseases of the studied children.

We thank the reviewer for this comment. We added in the text a new table as required.

  1. The list of references contains a lot of sources older than 5 years.

We acknowledge that the list of references includes many sources older than 5 years. However, given the rarity of Sotos syndrome and the lack of recent literature addressing heart disease in this syndrome, we had to rely on the limited recent case series available. Nonetheless, it was necessary to cite the extensive population studies conducted by the pioneers of the syndrome, despite their age